# Evaluation of Infective Endocarditis in Children: A 19-Year Retrospective Study in Taiwan

**DOI:** 10.3390/jcm12062298

**Published:** 2023-03-15

**Authors:** Shao-Ju Chien, Yi-Ju Tseng, Ying-Hua Huang, Hsi-Yun Liu, Yi-Hua Wu, Ling-Sai Chang, Yao-Hsu Yang, Ying-Jui Lin

**Affiliations:** 1Department of Pediatrics, Kaohsiung Chang Gung Memorial Hospital, Kaohsiung 83301, Taiwan; 2College of Medicine, Chang Gung University, Taoyuan 33302, Taiwan; 3Department of Early Childhood Care and Education, Cheng Shiu University, Kaohsiung 83347, Taiwan; 4Department of Computer Science, National Yang Ming Chiao Tung University, Hsinchu 30010, Taiwan; 5Health Information and Epidemiology Laboratory, Chiayi Chang Gung Memorial Hospital, Chiayi 61363, Taiwan; 6Department of Traditional Chinese Medicine, Chiayi Chang Gung Memorial Hospital, Chiayi 61363, Taiwan; 7School of Traditional Chinese Medicine, College of Medicine, Chang Gung University, Taoyuan 33302, Taiwan; 8Department of Respiratory Therapy, Kaohsiung Chang Gung Memorial Hospital and Chang Gung University College of Medicine, Kaohsiung 83301, Taiwan

**Keywords:** infective endocarditis, pediatrics, heart diseases, prognosis

## Abstract

Background: Infective endocarditis (IE) is an important cause of morbidity and mortality in pediatric patients with heart disease. Little literature has explored differences in the presentation of endocarditis in children with and without heart disease. This study aimed to compare the clinical outcomes and determine the risk of in-hospital death in the study population. Methods: Data were retrospectively collected from 2001 to 2019 from the Chang Gung Research Database (CGRD), which is the largest collection of multi-institutional electronic medical records in Taiwan. Children aged 0–20 years with IE were enrolled. We extracted and analyzed the demographic and clinical features, complications, microbiological information, and outcomes of each patient. Results: Of the 208 patients with IE, 114 had heart disease and 94 did not. Compared to those without heart disease, more streptococcal infections (19.3% vs. 2.1%, *p* < 0.001) and cardiac complications (29.8% vs. 6.4%, *p* < 0.001) were observed in patients with heart disease. Although patients with heart disease underwent valve surgery more frequently (43.9% vs. 8.5%, *p* < 0.001) and had longer hospital stays (28.5 vs. 12.5, *p* = 0.021), their mortality was lower than that of those without heart disease (3.5% vs. 10.6%, *p* = 0.041). Thrombocytopenia was independent risk factor for in-hospital mortality in pediatric patients with IE (OR = 6.56, 95% CI: 1.43–40.37). Conclusion: Among pediatric patients diagnosed with IE, microbiological and clinical features differed between those with and without heart disease. Platelet counts can be used as a risk factor for in-hospital mortality in pediatric patients with IE.

## 1. Introduction

Infective endocarditis (IE) occurs less often in children than in adults, accounting for approximately 1 in 1280 pediatric admissions per year [1,2]. However, the improved survival among children who are at risk of endocarditis, such as those with heart disease, has resulted in its increased incidence in recent decades [2]. With the change in IE epidemiology due to medical advances, the prevalence of rheumatic heart disease (RHD) has declined in developed countries, with an increase in association with congenital heart disease (CHD) in children. Ventricular septal defect (VSD), patent ductus arteriosus (PDA), aorta or aortic arch anomalies, and tetralogy of Fallot are common underlying conditions. An increasing proportion of corrective or palliative surgeries for CHD, with or without implanted patches, vascular grafts, or prosthetic cardiac valves, also contribute to IE incidence [3,4]. However, the clinical characteristics and outcomes of pediatric patients with IE and the differences in those with and without heart disease suffering from IE are limited. Therefore, this study aimed to describe and compare the clinical characteristics, outcomes, and major complications of IE between pediatric patients with and without heart disease. We also determined the risk factors of in-hospital death in the study population.

## 2. Materials and Methods

### 2.1. Study Design

In this retrospective cohort study, we collected data from 2001 to 2019 from the Chang Gung Research Database (CGRD), the largest multi-institutional database in Taiwan that includes de-identified electronic medical records from Chang Gung Memorial Hospitals (CGMHs) [5]. We collected data from seven branches of CGMHs that provide primary to tertiary medical care in Taiwan. Demographic patient data collected were the disease diagnoses, laboratory test results, management procedures, and prescribed medication from the inpatient departments. The diagnoses were identified using the International Classification of Diseases, Ninth and Tenth Revision codes (ICD-9 and ICD-10, respectively). Laboratory test results were recognized using the internally used codes in CGMHs and the procedures with the corresponding national health insurance (NHI) codes for valvular surgery. The study was approved by the Institutional Review Board (no. 201900978B0). The requirement for patient consent was waived because the identification numbers of all patients in the CGRD were encrypted to protect their privacy.

### 2.2. Definitions

Patients hospitalized with endocarditis diagnosis codes (ICD-9 codes 421.0, 421.1, 421.9, 424.9; ICD-10 codes I33, I38, I39) were retrieved [6] from the CGRD [6]. Echocardiographic reports with vegetation, abscess, aneurysm, pseudoaneurysm, perforation, fistula, and dehiscence were used to define infective endocarditis [7]. Predisposing heart conditions were defined as a history of having diagnosis codes of CHDs, cyanotic CHDs, pulmonary hypertension (Pul HTN), hypertrophic cardiomyopathy (HCM), Marfan syndrome, and RHD before admission. CHD lesions were further classified as VSD, atrial septal defect (ASD), endocardial cushion defect (ECD), PDA, and aorta/aortic arch lesions. Tetralogy of Fallot, univentricular heart, complete transposition complex, truncus arteriosus, and hypoplastic left heart syndrome with lesions most likely to be cyanotic at birth were grouped as cyanotic CHDs [6]. If one patient had multiple diagnoses, the diagnosis with the worst prognosis and highest complexity was established as the primary one [8]. For example, a patient with both ASD and TOF would be classified as having TOF. The involvement of different valves was also identified. In order to compare the clinical characteristics of pediatric patients with or without heart disease, the patients of above CHDs, cyanotic CHDs, or HCM were further grouped as heart disease (HD) for analysis. Complications that occurred during the admission period were retrieved and divided into three categories: (1) neurological (ischemic or hemorrhagic stroke), (2) respiratory (pulmonary embolism), and (3) cardiac (heart failure, myocardial infarction, or conduction disturbance) [9]. Appendix A provides a list of all ICD diagnoses or NHI procedure codes that were included in the study. To identify biomarkers to predict mortality, we analyzed the laboratory results of complete blood cell and differential counts that had been performed in 80% of the admitted patients. Leukocytosis and anemia were defined according to their normal ranges for the age group. A left shift indicated that more than 3% of immature neutrophils appeared in the blood. Thrombocytopenia was defined as a platelet count <150,000/mL.

### 2.3. Statistical Analysis

The characteristics of children with IE are described using absolute numbers, proportions, and median/interquartile ranges. Continuous variables are summarized as medians (interquartile ranges) and compared using the Kruskal–Wallis test. Categorical variables are presented as numbers (percentage) and were compared using Pearson’s χ^2^ test, or Fisher’s exact test if any expected cell count was <5 for the univariate analysis. Logistic regression with Firth’s correction, a solution to the problem of separation caused by rare events, was used to identify factors associated with mortality. We included age, sex, and variables with crude odds ratios (OR) that had a *p*-value of less than 0.2 in multiple variable analysis. From these analyses, we report the crude and adjusted ORs and their corresponding 95% confidence intervals (CI), estimated using the profile penalized likelihood method. Analyses were performed using R (version 4.1.3, the R Foundation for Statistical Computing, Vienna, Austria, www.r-project.org/, accessed on 13 August 2022). All statistical tests were two-sided, and *p* < 0.05 was considered statistically significant.

## 3. Results

### 3.1. Clinical Features and Major Outcomes

During the 19-year study period, 213,471 patients were diagnosed with IE in the emergency, outpatient, and inpatient departments. After excluding diagnoses made in the emergency and outpatient department or without image studies, 3455 cases of IE remained; 208 pediatric patients (aged 0–20 years old) were included in the final analysis.

Table 1 shows the demographic and clinical characteristics of the 208 pediatric patients with IE diagnostic codes. The median age was 13 years, and 57.8% of the patients were male (*n* = 118). Predisposing structural heart diseases (CHDs and cyanotic CHDs) were observed in 31.7% of patients (*n* = 66). Among the heart diseases, VSD was the most often encountered. The mitral valve was the most often involved. Several complications were observed, with cardiac complications being the most common (*n* = 40, 19.2%), including heart failure (*n* = 40, 19.2%), myocardial infarction (*n* = 1, 0.5%), and ventricular arrhythmias (*n* = 1, 0.5%). The second most common complication was pulmonary embolism (*n* = 15, 7.2%). Thirteen patients developed neurological complications, including ischemic and hemorrhagic stroke. Overall, 58 (27.9%) patients underwent valve surgery and 14 (6.7%) died.

### 3.2. Clinical Characteristics of the Subgroups

Approximately 27.9% of patients had positive blood cultures. Staphylococcus species (*n* = 33, 15.9%) was the most frequent contributing pathogen found in all the study participants, followed by Streptococcus species (*n* = 24, 11.5%). The clinical characteristics of the patients with and without HD are listed in Table 2. Streptococci were the most frequently identified pathogens in the HD group (*n* = 22; 19.3%). There were two Pseudomonas and Candida infections in the non-HD group. Staphylococci (*n* = 17, 18.1%) were the most common causative pathogens identified in the non-HD group, especially *Staphylococcus aureus*. Complications, especially cardiac problems, were more commonly found in the HD group than in the non-HD group (39.5% vs. 16%, *p* = 0.0002). No significant difference was observed in the frequency of neurological complications and pulmonary embolisms between the two groups. Although patients with HD underwent valve surgery more frequently (43.9% vs. 8.5%, *p* < 0.001) and had longer hospital stays (27.4 vs. 22.3 days, *p* = 0.021), non-HD patients had a significantly higher in-hospital mortality rate (10.6% vs. 3.5%, *p* = 0.0411).

### 3.3. Risk Factors for In-Hospital Mortality

Anemia and leukocytosis were common in IE; thrombocytopenia accounted for one-third (33.1%) of the patients. Fourteen patients (6.7%) died during hospitalization. The risk factors associated with in-hospital mortality were further analyzed in 166 patients with complete laboratory data (Table 3). There were more left-shifted immature neutrophils and thrombocytopenia in the mortality group than in the survival group (66.7% vs. 20.1%, *p* = 0.001 and 83.3% vs. 29.2%, *p* < 0.001, respectively). In thrombocytopenia episodes, the mean platelet count was 71.4 × 10^3^ (SD 43.4 × 10^3^), whereas, in those without thrombocytopenia, it was 269.2 × 10^3^ (SD 99.5 × 10^3^). Although streptococci infection, cardiac complications, valvular surgery, and longer hospital stay were more common in pediatric patients with HD than in those without HD suffering from IE, the only significant risk factor for in-hospital death was thrombocytopenia (adjusted OR: 6.56; 95% CI: 1.43–40.37; *p* = 0.015) in the multivariable analysis.

## 4. Discussion

The results of this study exhibited significant differences in the causative microbiological features, clinical characteristics, and outcomes between pediatric patients with and without HD suffering from IE. The proportion of Streptococci species and cardiac complication rates were significantly greater in the HD group than in the non-HD group. Although more valve surgeries and longer hospital stays were encountered in the HD group, the mortality rate was lower than that in the non-HD group. In this study, thrombocytopenia was independently associated with overall in-hospital mortality.

In our study, there were certain negative culture results. The different incidences of IE with negative blood culture in different countries resulted from the epidemiology of IE, the diagnostic criteria accepted, and the diagnostic technique [10,11]. Despite improvements in blood culture media and the introduction of automated growth detection systems, blood-culture-negative IE still represents 5–69.7% of all cases of endocarditis [12]. The main clinical scenario is early empiric antibiotic treatment started prior to blood sampling before the patient is diagnosed with or suspected of having IE in many critical clinical situations [10,13]. Due to the observational nature of the analysis, no detailed clinical information was captured regarding the exact timing of antimicrobial therapy with respect to IE diagnosis and culture methods.

Blood-culture-negative IE poses a significant challenge in clinical determination of the best specific and effective antibiotic choice for this life-threatening infection. Although blood and valve tissue cultures are the gold standard for IE pathogens’ detection, they are very time-consuming and unfeasible for fastidious or intracellular microorganisms [14]. Many modified criteria and diagnostic techniques, such as imaging, have been introduced in an effort to improve the diagnosis of pathologically proven native valve blood-culture-negative IE [15]. The value of echocardiography in the diagnosis of patients with culture-negative endocarditis is great since it improves sensitivity and specificity [16,17] and has proven to be a major criterion for endocarditis in 78–98.5% of definite IE cases [15,17]. Although there are no major microbiological criteria, blood-culture-negative endocarditis has been shown to be associated with younger age, shorter duration of symptoms at time of presentation [18], more a frequent need for surgical repair [19], a greater number of intra- and postoperative complications, and consequently, worse outcomes and disease courses [20]. Due to these important issues, we did not exclude patients with blood-culture-negative IE, and we used echocardiographic evidence as a major contributing factor in the diagnosis of IE cases.

Although IE is a rare condition, it remains a considerable concern in patients with underlying cardiac conditions, whether they are unrepaired, palliated, or corrected [2,21]. Even with modern medicine and improved survival of patients with CHD, it remains a major predisposing risk factor for IE in childhood and adolescence [22,23]. In a Canadian national study, the most common CHD with IE was cyanotic CHD lesions, followed by ASD and VSD [6]. The IE risk varies markedly with the complexity of lesions among patients with CHD; patients with more complex lesions or severe defects are at higher risk of IE. However, persistent turbulent flow after cardiac operation, even in simple CHD, can cause endocardial damage, allowing for adhesion of microorganisms following transient bacteremia [24]. In our study, we were unable to determine whether patients had completed surgical repair before IE or if there were residual cardiac lesions after surgery. Furthermore, the IE involved more left-sided mitral and aortic valves. In developed countries, there is an increasing temporal trend in mitral valve prolapse among patients suffering from IE with underlying HD [25,26]. The relatively high IE risk observed in children with left-sided lesions is in accordance with several research articles that have emphasized the IE risk in patients with left ventricular outflow obstruction lesions [6]. Isolated secundum ASD has negligible IE risk, but the IE risk in ASD is higher than that in the general population in the Quebec CHD database and CONCOR database of adults with CHD [6,27]. These patients may have had coexistent lesions that rendered them vulnerable to IE, such as valvular lesions (particularly regarding the mitral valve) or primum ASD that may have been misclassified and not contained in the ECD group [6]. The infrequency of IE prevented the investigation of individual CHD lesions, and the fewer IE events per specific CHD lesion prohibited any statistical analysis.

Patients with IE have a broad spectrum of complications. Cardiac complications were the principal cause of morbidity, and neurological or embolic complications were significantly fewer in our study than in the previous studies [3,28]. The highly variable clinical presentation of IE may reflect the variable causative microorganisms and underlying cardiac conditions. Patients with HD suffering from IE had significantly different clinical characteristics from those without HD patients. Our study showed that the most common causative pathogen of IE was Staphylococcus species (15.9%), followed by Streptococcus species (11.5%). However, among children with HD, the most frequent causative microorganism was *Streptococcus*, rather than *Staphylococcus*. These data concur with those of other studies, in which the frequency distribution of *Staphylococcus aureus* in IE in North America was significantly increased [29]. However, *Staphylococcus* distribution was less in patients with HD [30,31]. *S. aureus* in IE has a higher rate of complications, particularly neurological complications, and mortality when compared with other pathogens [3,32]. Although patients with underlying HD are known to have a worse prognosis [31], our decreased morbidity and mortality rate might be partially due to the lower percentage of *S. aureus* as the underlying pathogen. Among our pediatric patients, 27.9% underwent surgery. Management of progressive valvular damage and the consequential heart failure with medical therapy alone is often ineffective [1]. Cardiovascular surgery may be lifesaving in patients with IE if they have progressive cardiac failure, valvular obstruction, persistent bacteremia even under appropriate antibiotic therapy, or significant embolic events [33].

Our overall mortality rate was 6.7%, and thrombocytopenia was common in pediatric IE (33.1%) at a rate similar to that of 32.8% in an adult study [34]. Thrombocytopenia was the only laboratory parameter that was associated with mortality. A variety of nonspecific laboratory findings were noted in children by [1], such as anemia, leukocytosis, or immature forms of white blood cells, and their proportion was higher in our patients with mortality. However, none were independent predictors of mortality in the multivariable analysis. Thrombocytopenia was the only risk factor, superseding HD, involving valves, complications, or the need for surgery, which was associated with mortality. Patients with thrombocytopenia had a higher mortality rate than patients without thrombocytopenia. Although only a few retrospective studies have identified thrombocytopenia as a risk factor for death in adult patients with IE [35,36], it is a known risk factor for death in sepsis. Furthermore, patients with thrombocytopenia are more likely to develop stroke and persistent signs of infection and septic shock [37]. Thrombocytopenia has been included in several severity scores for sepsis and multiple organ dysfunction [38,39]. A prospective study in three tertiary care university hospitals was conducted by C. Ferrera Duran et al., who stated that thrombocytopenia at admission should be used to identify the patients with native valve IE at risk of complicated in-hospital complications and mortality. Thrombocytopenia could be included in IE guidelines as an early marker for risk stratification and predictor of a poor outcome [34]. Our study is the first pediatric series to identify thrombocytopenia as the laboratory parameter associated with mortality, though it echoes previous literature. Therefore, thrombocytopenia could be considered for integration into the triad of clinical, microbiological, and echocardiographic issues [40] that predict pediatric patient outcomes in IE. Platelets play the primary role in thrombosis, hemostasis, inflammation, and infection. They can directly interact with pathogens and immune cells via their surface receptors [41]. In endocarditis pathogenesis, platelets play an important role in the local host defense against endovascular infections [42]. In addition, they are sensitive monitors of the systemic host response to sepsis and other severe clinical symptoms [36]. In patients with sepsis, thrombocytopenia may be induced by impaired platelet production or increased immune and non-immune destruction/consumption of platelets [43]. A synergistic interaction between thrombocytopenia and *Streptococcus aureus* has also been observed to have an effect on mortality risk [34,44]. In addition, drug-induced thrombocytopenia might also represent a minority of cases among thrombocytopenic children. However, we could not definitively elucidate these in our patient cohort. Further bone marrow aspirates or autoantibody-specific assays may help investigate possible mechanisms of thrombocytopenia.

## 5. Limitations

We measured IE from hospital administrative records, similar to several large-scale research studies designed to acquire absolute measures of IE frequency, underlying diseases, complications, and mortality [6,22]. However, the study design has potentially referral bias because all the participants were cared for by primary to tertiary medical hospitals of CGMHs. Furthermore, we conducted an analysis of a sub-cohort with complete laboratory data on mortality, for which there could have been a selection bias. Beyond that, some lack of data prevented the assessment of and adjustment for potential confounding factors across patient subpopulations. Additionally, an inherent limitation of such an administration-based study was that detailed clinical criteria to establish IE diagnosis were not available. The lack of proper validation studies for ICD codes for IE and misclassification of IE could result in overestimation or underestimation of the cumulative IE incidence. This would have led to bias toward the unfounded value in the predictor analysis. Furthermore, the retrospective design of the present study may have overlooked some clinical features, inconsistencies in patient care, and discrepancies in patients’ underlying background information. Within the CHD group, our data did not allow for a distinction between native anatomy without surgical repair and postoperative status. Moreover, the long-term survival and emergence of further complications after hospital discharge were not identified. Further refinement of more subtle clinical data and details about patient care and antimicrobial therapy are needed for more accurate and extensive determination of the risk of IE. Nevertheless, a unique feature and strength of the study was the use of the only large, long-term, multi-institutional inpatient database currently available in Taiwan. Furthermore, we were able to compare the manifestation of IE in the HD and non-HD pediatric patients and provide relevant information. In the future, further prospective studies or randomized trials could be conducted to generate further validation and evidence to support the management of IE.

In conclusion, although patients with HD had a higher proportion of IE and cardiac complications than those without HD, HD itself was not an independent risk factor for in-hospital mortality in our study. Thrombocytopenia during admission could be used for risk stratification in pediatric patients with IE to identify increased mortality. If antiplatelet agents are considered as adjunctive therapy for embolic complications of IE, clinicians should be cautious of coexistent thrombocytopenia as it might increase the risk of bleeding. In addition, patients with thrombocytopenia may warrant more intensive monitoring and dedicated adjustments to treatment, and, where relevant, surgery should be considered.

## Figures and Tables

**Table 1 jcm-12-02298-t001:** General characteristics of 208 pediatric patients with infective endocarditis.

	N (*n* (%))	Age (Median (IQR))
Patient, *n*	208	13 (4, 18)
Sex, *n* (%)		
Male	118 (56.7)	13 (3, 18)
Female	90 (43.3)	14 (6–19)
CHD	40 (19.2)	7 (2, 15)
ASD	1 (0.5)	5
PDA	3 (1.4)	17 (3, 19)
VSD	28 (13.5)	9 (3, 15)
ECD	7 (3.4)	1 (0, 8)
Ao arch	9 (4.3)	6 (1, 12)
Cyanotic CHD	26 (12.5)	4 (0, 15)
Pul HTN	10 (4.8)	14 (4, 15)
HCM	3 (1.4)	10 (3, 18)
Marfan	1 (0.5)	15
RHD	6 (2.9)	17 (12, 19)
Valve involved		
Mitral	48 (23.1)	17 (14, 19)
Aortic	16 (7.7)	13 (8, 17)
Mitral and aortic	3 (1.4)	9 (8, 19)
Pulmonary	19 (9.1)	4 (0, 12)
Tricuspid	8 (3.8)	15 (9, 16)
Prosthetic valve	4 (1.9)	13 (10, 18)
Complications		
Neurologic	13 (6.3)	17 (14, 19)
Cardiac	40 (19.2)	13 (3, 18)
Pulmonary embolism	15 (7.2)	15 (8, 20)
Valve surgery	58 (27.9)	14 (5, 19)
Mortality	14 (6.7)	9 (0, 19)

CHD, congenital heart disease; VSD, ventricular septal defect; ASD, atrial septal defect; ECD, endocardial cushion defect; PDA, patent ductus arteriosus; Ao arch, aorta/aortic arch lesions; Pul HTN, pulmonary hypertension; HCM, hypertrophic cardiomyopathy; RHD, rheumatic heart disease.

**Table 2 jcm-12-02298-t002:** Comparison of the clinical characteristics of pediatric patients with and without heart disease suffering from infective endocarditis.

Characteristics	Overall (*n* = 208)	Heart Disease ^1^ (*n* = 114)	Non-Heart Disease (*n* = 94)	*p*-Value
GENERAL				
Age, y (median [IQR])	13 (4, 18)	13 (4, 18)	11.5 (4, 18)	0.7671
Male, *n* (%)	118 (56.7)	66 (57.9)	52 (55.3)	0.7091
MICROBIOLOGY	58 (27.9)	37 (32.5)	21 (22.3)	0.1054
Streptococci, *n* (%)	24 (11.5)	22 (19.3)	2 (2.1)	0.0001
Staphylococci, *n* (%)	33 (15.9)	16 (14.0)	17 (18.1)	0.4262
Staphylococcus aureus, *n* (%)	27 (13.0)	14 (12.3)	13 (13.8)	0.7408
Pseudomonas, *n* (%)	2 (1.0)	0 (0)	2 (2.1)	0.203
Candida, *n* (%)	2 (1.0)	0 (0)	2 (2.1)	0.203
COMPLICATIONS	60 (28.9)	45 (39.5)	15 (16)	0.0002
Neurologic ^2^, *n* (%)	13 (6.3)	10 (8.8)	3 (3.2)	0.098
Cardiac ^3^, *n* (%)	40 (19.2)	34 (29.8)	6 (6.4)	<0.0001
PE, *n* (%)	15 (7.2)	8 (7.0)	7 (7.4)	0.9052
VALVE SURGERY, *n* (%)	58 (27.9)	50 (43.9)	8 (8.5)	<0.0001
MORTALITY, *n* (%)	14 (6.7)	4 (3.5)	10 (10.6)	0.0411
Hospital stay (median (IQR))	20 (6, 40.5)	28.5 (8.25, 42.75)	12.5 (5, 36.75)	0.021

Heart disease ^1^ = congenital heart disease (CHD) + cyanotic congenital heart disease + hypertrophic cardiomyopathy (HCM). Neurologic ^2^ = ischemic + hemorrhagic stroke. Cardiac ^3^ = heart failure + myocardial infarction + ventricular arrhythmia.

**Table 3 jcm-12-02298-t003:** Predictors for mortality in 166 patients with IE and complete laboratory data for blood cell counts.

Risk Factors	Overall (*n* = 166)	Recovered (*n* = 154)	Died (*n* = 12)	*p*-Value	Univariable Analysis	Multivariable Analysis
					OR (95% CI ^6^)	aOR (95% CI)
Age (median [IQR])	13.19 [4.78, 18.67]	13.19 [4.95, 18.72]	8.91 [0.26, 18.36]	0.201	0.96 (0.88–1.04)	0.92 (0.83–1.01)
Male, *n* (%)	76 (45.8)	70 (45.5)	6 (50.0)	0.773	1.2 (0.36–4.00)	1.32 (0.36–4.9)
Heart disease ^1^, *n* (%)	91 (54.8)	87 (56.5)	4 (33.3)	0.141	0.41 (0.11–1.29)	0.5 (0.12–1.8)
Microbiology, *n* (%)	57 (34.3)	53 (34.4)	4 (33.3)	1	1 (0.28–3.19)	- ^7^
Leukocytosis, *n* (%)	114 (68.7)	104 (67.5)	10 (83.3)	0.344	2.03 (0.56–10.82)	-
Left shift, *n* (%)	39 (23.5)	31 (20.1)	8 (66.7)	0.001	7.41 (2.29–27.17)	2.98 (0.78–12.67)
Anemia, *n* (%)	131 (78.9)	121 (78.6)	10 (83.3)	1	1.16 (0.32–6.22)	-
Thrombocytopenia, *n* (%)	55 (33.1)	45 (29.2)	10 (83.3)	<0.001	10.11 (2.79–53.99)	6.56 (1.43–40.37)
Neurologic complications ^2^, *n* (%)	12 (7.2)	10 (6.5)	2 (16.7)	0.464	3.28 (0.58–13.45)	3.52 (0.49–19.71)
Cardiac complications ^3^, *n* (%)	36 (21.7)	34 (22.1)	2 (16.7)	0.941	0.83 (0.15–3.05)	-
Pulmonary embolism, *n* (%)	12 (7.2)	12 (7.8)	0 (0.0)	0.671	0.46 (0–3.86)	-
Lt-sided valve ^4^	51 (30.7)	49 (31.8)	2 (16.7)	0.441	0.51 (0.1–1.84)	-
Rt-sided valve ^5^	20 (12.0)	18 (11.7)	2 (16.7)	0.96	1.76 (0.32–6.72)	-
Valve surgery, *n* (%)	52 (31.3)	50 (32.5)	2 (16.7)	0.416	0.49 (0.09–1.78)	-

Heart disease ^1^ = congenital heart disease (CHD) + cyanotic congenital heart disease + hypertrophic cardiomyopathy (HCM). Neurologic ^2^ = ischemic + hemorrhagic stroke. Cardiac ^3^ = heart failure + myocardial infarction + ventricular arrhythmia. Lt-sided valve ^4^ = mitral valve, aortic valve, and aortic and mitral valves. Rt-sided valve ^5^ = pulmonic valve, infundibular stenosis, tricuspid valve, and Ebstein anomaly. CI ^6^, confidence interval. - ^7^, variables were not included in the multivariable analysis.

## Data Availability

The data is unavailable due to CGRD restrictions.

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
