# Peer review of "Evaluation of Infective Endocarditis in Children: A 19-Year Retrospective Study in Taiwan"

_jcm, 2023, doi:10.3390/jcm12062298_

Round 1

Reviewer 1 Report

I would like to congratulate the authors on a good effort. The current study describes paediatric IE.

Comment 1: It is well known that predisposing valve conditions are a risk factor for the coccurrence of Endocarditis. What were the causes of endocarditis in the non Heart disease group ?

Comment 2: Line 74-76: Check Formatting.

Comment 3: Line 134-138: Pathogen spectrum: The authors should present this data in more details. Are there cases of BCNIE ? Which Streptococci caused endocarditis ?

Comment 4: Discussion:  Thrombocytopenia is a known indicator for poor outcomes in Endocarditis.  This should be discussed with reference to actual studies and clinical results.

C. Ferrera Duran et al. Thrombocytopenia, a new marker of bad prognosis in patients with infective endocarditis, European Heart Journal, Volume 34, Issue suppl_1, 1 August 2013,P4773, https://doi.org/10.1093/eurheartj/eht310.P4773

Comment 5: The authors describe a 19 year experience. It would be interesting to know about the long term survival, recurrence and complications.

Comment 6 : Discussion: Please describe the pathogen spectrum and how it compares to adult IE. 

Comment 7: It would be interesting to know about the foci for infective endocarditis and its etiology. 

Author Response

as attached file

Reviewer 2 Report

We appreciate the opportunity to review this manuscript.

The aim of this study was to compare clinical outcomes and determine the risk of in-hospital death in pediatric patients with infective endocarditis (IE) with and without heart disease (HD). Data were retrospectively collected from 208 children aged 0-20 years with IE in Taiwan from 2001 to 2019. Patients with HD had more streptococcal infections and cardiac complications, underwent valve surgery more frequently, and had longer hospital stays but lower mortality compared with those without HD. Thrombocytopenia was identified as an independent risk factor for in-hospital mortality in pediatric patients with IE.

This is a very important issue, because the knowledge about IE in children, and even more so in children with congenital HD, is limited. The authors rightly claim that they performed a large database analysis of children with C/HD suffering from IE for the first time. The analysis, within the limitations of a retrospective analysis, is thorough and the results have been extensively discussed.

However, there are major concerns regarding the presentation and conclusions of the study, as follows:

1. In Table 3 - the reader may get the impression that all parameters were forced into the multivariable logistic regression analysis. Therefore, the statistical power of the analysis would suffer, which would be reflected in extremely wide confidence intervals, e.g. for thrombocytopenia with 95%-CI: 2.75-190.18. As a result, the conclusion of the analysis may be spurious. Please clarify.

2. Table 2: Does neurological = ischemic + hemor. mean both or only one of them? Same for cardiac. Please clarify.

3. Somehow the different numbers of CHD (n=66) in Table 1 and Table 2 (n=114) are not clearly explained. It is not clearly formulated whether additional cardiac pathologies have been introduced in Table 2. Please clarify.

4. Table 3: T-test? also for categorical variables? How is OR presented for continuous variables, per unit increase or interquartile range or per standard deviation? Please clarify.

5. Another major concern remains the analysis of the sub-cohort with complete laboratory data on mortality. Is this subcohort representative of the entire cohort? Could there be a bias in selecting only patients with complete laboratory data and could this bias lead to point 6?

6. The authors' final conclusion that thrombocytopenia is the "sole" predictor of mortality in children with IE may be too enthusiastic. The conclusion should be formulated more cautiously. Indeed, the authors mention the limitations of this retrospective study, but should clearly state that further prospective validation studies are needed for their claim.

Minor comments:

10. P6 L 199-201: The text is not clear, what was the intention of the authors, are they referring to reference #6 or to their own study?

11. P6 L 209 - 213 and P7 L 259-260: some text passages are not clearly formulated.

Author Response

as attached file

Round 2

Reviewer 1 Report

The authors have done their best to answer the querries.

Author Response

Manuscript Number: JCM-2254601

Manuscript Title: Evaluation of infective endocarditis in children: A 19-year retrospective study in Taiwan

Response to Reviewer #1

Comment: The authors have done their best to answer the querries.

Response: Thank you for your comprehensive review and valuable comments.

Reviewer 2 Report

Thank you for the opportunity to review the revised version of this manuscript.

This paper addresses a very important topic, as knowledge about IE in children, and even more so in children with congenital HD, is limited.

The authors have responded to most of the comments, but two issues remain:

1. Regarding the multivariable regression analysis.

The authors have correctly implemented logistic regression with Firth's correction and penalised the likelihood method of estimation in the revised version of the manuscript due to the limited number of events (n = 14) in a cohort of 166 patients. However, it appears that the authors performed a multivariable regression analysis with 14 predictors, which would imply one predictor for one event. Even if one relaxes the rule as outlined by Vittinghoff et al. (2007, American Journal of Epidemiology) and would consider a higher predictor prevalence of ~50% of the cohort, the actual approach of the authors would decrease the coverage of the confidence interval, increase the type I error and the relative bias of the estimates. Thus, the control of confounders would not be covered and the conclusion might be spurious.Please clarify.

2. The following sentence (on page 8, lines 278 to 279 of the revised manuscript) "It is time for thrombocytopenia to be included in IE guidelines as an early marker for risk stratification and predictor of poor outcome [34]" should be rephrased (toned down), as Ferrera et al [reference #34 of the revised manuscript] wrote this call more cautiously as follows: "Thrombocytopenia could be added to the triad of clinical, microbiological, and echocardiographic parameters that can rapidly predict patient outcome in IE on admission, especially in native valve IE. Although Ferrera et al. conducted a prospective study, they also noted the limitations of their study, including referral bias, single measurement of thrombocytopenia, and possible drug-induced thrombocytopenia.

Author Response

as attached file. Thank you!
